# Analysis of Cytokine and Chemokine Level in Tear Film in Keratoconus Patients before and after Corneal Cross-Linking (CXL) Treatment

**DOI:** 10.3390/ijms25021052

**Published:** 2024-01-15

**Authors:** Magdalena Krok, Ewa Wróblewska-Czajka, Olga Łach-Wojnarowicz, Joanna Bronikowska, Zenon P. Czuba, Edward Wylęgała, Dariusz Dobrowolski

**Affiliations:** 1Chair and Clinical Department of Ophthalmology, Faculty of Medical Sciences in Zabrze, Medical University of Silesia in Katowice, Panewnicka 65 Street, 40-760 Katowice, Polandewaw8@wp.pl (E.W.-C.); wylegala@gmail.com (E.W.); dardobmd@wp.pl (D.D.); 2Ophthalmology of Department, District Railway Hospital, 65 Panewnicka Street, 40-760 Katowice, Poland; 3Department of Microbiology and Immunology, Faculty of Medical Sciences in Zabrze, Medical University of Silesia, 40-055 Katowice, Poland; jbronikowska@sum.edu.pl (J.B.); zczuba@sum.edu.pl (Z.P.C.)

**Keywords:** keratoconus, corneal cross-linking, cytokines, immunology, cornea

## Abstract

Keratoconus (KC) is a degenerative corneal disorder whose aetiology remains unknown. The aim of our study was to analyse the expressions of cytokines and chemokines in KC patients before and after specified time intervals after corneal cross-linking (CXL) treatment to better understand the molecular mechanisms occurring before and after CXL in KC patients process of corneal regeneration.; Tear samples were gathered from 52 participants immediately after the CXL procedure and during the 12-month follow-up period. All patients underwent a detailed ophthalmological examination and tear samples were collected before and after CXL at regular intervals: 1 day before and after the surgery, at the day 7 visit, and at 1, 3, 6, 9, and 12 months after CXL. The control group consisted of 20 healthy people. 10 patients were women (50%) and 10 were men (50%). The mean age was 30 ± 3 years of age. Tear analysis was performed using the Bio-Plex 3D Suspension Array System. Corneal topography parameters measured by Scheimpflug Camera included: keratometry values (Ks, Kf), PI-Apex, PI-Thinnest, Cylinder.; All the 12 months post-op values of the KC patients’ topographic measurements were significantly lower than the pre-op. As for the tear cytokine levels comparison between the patient and control groups, cytokine levels of TNF-α, IL-6, and CXCL-10, among others, were detected in lower amounts in the KC group. The pre-op level of IL-6 exhibited a significant increase the day after CXL, whereas comparing the day after the procedure to 12 months after CXL, this showed a significant decrease. Both TNF-α and IL-1 showed a significant decrease compared to the day before and after CXL. We observed significantly higher levels of IL-1β, IL-10, IFN-γ and TNF-α in moderate and severe keratoconus than in mild keratoconus (*p* < 0.05). We also demonstrated a statistically significant positive correlation between both pre-op and 12 months after CXL TNF-α, IFN-γ, IL-6 and Ks and Kf values (*p* < 0.05, r > 0); Alterations of inflammatory mediators in tear fluid after CXL link with topographic changes and may contribute to the development and progression of KC.

## 1. Introduction

Keratoconus (KC) is a degenerative corneal disorder usually occurring in men’s late childhood to early adulthood with the estimated prevalence in the general population 54 per 100,000. However, these data may be underestimated since the most recent nation-wide study performed in the Netherlands revealed a five to ten times higher prevalence of keratoconus than previously documented (265 per 100,000). [1] Keratoconus is characterised by progressive thinning of the cornea which can lead to a cone-shaped protrusion of the cornea and subsequently to myopia, irregular astigmatism and severe visual impairment. Keratoconus treatment depends on the degree of severity and progression. In the early stages spectacles or soft contact lenses (CL) may be sufficient to improve visual acuity. In case of unsatisfactory vision, gas-permeable rigid CL should be recommended. A wide range of contact lenses on the market allows us to offer alternative types, from hybrid lens and piggy-back to corneoscleral and scleral lens [2]. Nonetheless, in patients with a documented clinical progression we should perform corneal cross-linking (CXL), the aim of which is to stiffen the cornea by means of a photochemical reaction that occurs after removal of the central 6–7 mm of corneal epithelium and applying riboflavin (B2 vitamin isomer) followed by irradiation with UVA light at 370 nm [3]. The severe cases may require corneal surgery (intrastromal corneal ring implantation, refractive surgery, keratoplasty).

Despite considerable research, the exact aetiology of the disease remains unknown. Previous research results define KC as a non-inflammatory disease and suggest a multifactorial aetiology, with an indication of genetic, environmental and biomechanical risk factors. Several studies have shown that the imbalance between cell adhesion molecules, and matrix metalloproteinases (MMPs) and pro- and anti-inflammatory cytokines contained in tears of KC performs a crucial role in the disrupted corneal homeostasis [4,5,6,7,8,9].

The tear film is a unique fluid layer that covers the outer mucosal surfaces of the eye. On account of its aqueous/mucin phase it provides not only antimicrobial defence but also makes a contribution to wound healing and inflammatory responses. As reported by recent studies, the tear film could play a homeostatic role due to its pro- and anti-inflammatory cytokines content. Hence, the tears cytokine imbalance was proven to occur in patients with ocular surface diseases, such as dry eye syndrome (DES) [10,11,12,13,14], keratoconus (KC) [8,9,15,16,17,18,19], primary open-angle glaucoma, and proliferative diabetic retinopathy, among others [20,21]. Ongoing research sheds new light on understanding the pathogenesis and therefore the therapy of KC. The cytokines selected for analysis act as immunomodulators. The main pro-inflammatory cytokines are IL-1, IL-6, IL-8, IL-15 and TNF-α, IFN-γ. The initiation of inflammation depends on this group of cytokines. Il-13 is a mediator of allergy-related inflammation. Chemokines are also essential in inflammatory processes because they act chemotaxis through chemokine receptors and influence the migration of immune system cells in tissues [22,23]. The aim of our study was to analyse the expressions of cytokines and chemokines in KC patients before CXL treatment and examine the potential changes in the concentration of these biomarkers at specified time intervals after the intervention [24].

## 2. Results

A total of 52 eyes of 52 patients were successfully monitored for one year. A total of 13 patients were women (25%) and 39 were men (75%). The mean age was 24.35 ± 7.88 years. An amount of 20 patients constituted the control group, where 10 patients were women (50%) and 10 were men (50%). The mean age was 30 ± 3 years of age.

### 2.1. Values of Topographic Measurements

Table 1 summarises the topographic measurements of KC patients and comparison of pre- and post-CXL values. All the 12 months post-op values were significantly lower than the pre-op. Ks (49.8 ± 1.1 vs. 48.7 ± 1.1, respectively, and *p* = 0.001), Kf (46.3 ± 1.0 vs. 45.4 ± 1.0, respectively, and *p* < 0.001), PI-Apex (482.1 ± 12.0 vs. 461.6 ± 12.3, respectively, and *p* = 0.002), PI-Thinnest (450.8 ± 10.9 vs. 431.4 ± 12.0, respectively, and *p* = 0.011), though the CYL post-op value were significantly lower than 7 days after CXL (4.0 ± 0.5 vs. 3.3 ± 0.5, respectively, and *p* ≤ 0.001).

Topographic measurements of control group include Ks (43.98 ± 1.54), Kf (42.98 ± 0.87), CYL (1.02 ± 0.45), PI-Apex (544.6 ± 28.4) and PI-Thinnest (528.0 ± 41.9).

### 2.2. Cytokine Measurement Values

As for the tear cytokine levels comparison between the patient and control groups, cytokine levels of TNF-α, IL-6, CXCL-10, NGF-β, IFN-γ, CCL-3, CCL4, FGF-basic, IL-15, IL-1β, IL-13, IL-12 were detected at lower levels in the KC group (*p* < 0.05). When considering the cytokine levels over the course of this study, the pre-op level of IL-6 exhibited a significant increase the day after CXL (31.7 ± 96.1 vs. 296.3 ± 96.0, respectively, *p* = 0.004). Comparison of IL-6 levels the day after the procedure and 12 months after the CXL showed a significant decrease (296.3 ± 96.0 vs. 18.0 ± 129.8, respectively, *p* = 0.004). Both TNF-α (15.5 ± 3.7 vs. 9.6 ± 3.7, respectively, *p* = 0.03) and IL-1 (23.8 ± 7.0 vs. 7.0 ± 13.9, respectively, *p* = 0.009) revealed a significant decrease compared the day before and after CXL (Figure 1).

### 2.3. Correlation of Cytokines with Cornea Topographic Parameters

By correlating corneal and immunological data, we observed significantly higher levels of IL-1β, IL-10, IFN-γ and TNF-α in moderate and severe keratoconus than in mild keratoconus (*p* < 0.05), but not exceeding the values in the control group. We also demonstrated a statistically significant positive correlation between both pre-op and 12 months after CXL TNF-α, IFN-γ, IL-6 and Ks and Kf values (*p* < 0.05, r > 0). Table 2.

## 3. Discussion

The aetiology of keratoconus remains unexplained. We also do not know the factors influencing the progression of the disease. Keratoconus is still considered a non-inflammatory disease, despite several articles claiming otherwise [4,7,8,9,25,26,27,28,29]. Among the many functions of the tear film, its most important role is the protection of the eye’s surface. This involves the presence of various anti-inflammatory proteins, cytokines, and chemokines. Since the tear film is in direct contact with the eye’s surface, analysing its composition may help identify biomarkers for faster diagnosis and targeted therapy of eye surface diseases [24]. Currently, it has been proven that the composition of the tear film changes in some chronic eye surface diseases, such as DES [10,11,12,13,14].

Based on our knowledge, this study features the largest patient group to date, examining cytokine and chemokine levels immediately after the CXL procedure in the tear film of individuals with keratoconus and tracking these concentrations over a one-year period immediately after the CXL procedure. From what we know, there is no study with a longer follow-up period in this group of patients.

Despite studies on tear factors in patients with keratoconus after CXL procedure being the subject of several research papers [19,21,29,30,31,32], this is the first study with such a large research group analysing cytokine and chemokine levels immediately after CXL procedure in KC patients. Our study indicates a disruption of the immune homeostasis of the tear film in patients with keratoconus after CXL. Perhaps labelling keratoconus as a non-inflammatory disease is inappropriate, considering the statistically significant reduction in IL-6 concentration in each of our studied time periods: after seven days, after a month, and after 3, 6, 9 and 12 months. This is in line with earlier observations of Balasubramanian et al. [30], who paid special attention to the significant increase in IL-6 in patients with KC compared to the CXL-treated group and Kolozsvári et al. [19] who also noted a decrease in IL-6 twelve months after CXL. However, compared to the latter, the decrease in IL-8 in our study was not statistically significant.

Furthermore, in our study, we also observed a statistically significant decrease in IL-1β and IL-1R one day after CXL. These proteins are the most thoroughly studied cytokines, both because they were discovered first and because they have strong pro-inflammatory effects. Therefore, the reduction in their concentrations after the procedure has a beneficial effect on the corneal condition [19].

The main pro-inflammatory cytokines that initiate the inflammatory response, in addition to the mentioned IL-1 and IL-6, also include TNF-α. Its concentration also significantly decreases one day after the procedure. The trend in the subsequent periods we investigated continues to be a decreasing one, although not statistically significant. The aforementioned pro-inflammatory cytokines are primarily produced by macrophages and monocytes and stimulate the maturation of antigen-presenting cells (APC). IL-1β, in addition to its pro-inflammatory nature, is also involved in cell apoptosis and sensitivity to pain [33,34]. An interesting fact is that these cytokines undergo strong expression in other ocular diseases, such as DED. [35]

The transient worsening of the local corneal condition after CXL and the temporary change in inflammatory mediators coincide in time with postoperative apoptosis, clearing, and the growth of new epithelium, as well as the reoccupation of keratocytes and the synthesis of new collagen [36,37,38]. The UVA used in the CXL technique induces the cell death of outer corneal keratocytes, which is a necessary process to scaffold new tissue in their place [39]. When cells die due to necrosis, a strong inflammatory response is initiated, leading to the release of various inflammatory mediators, including cytokines and chemokines. On the first day after CXL, the extremely elevated concentration of IL-6 detected in our study significantly contributes to the healing process of the corneal epithelium and the recruitment of inflammatory cells to the corneal stroma, preventing tissue damage from excessive inflammation by removing damaged cells [40]. After CXL, all layers of the cornea regenerate quickly, and even epithelial regrowth is completed within four days. After corneal re-epithelialisation, remodelling, and reorganisation, new keratocytes migrate to the central area within a few months after CXL. The gradual repopulation of the corneal stroma begins between the second and third month after the intervention, usually concluding within six months [38].

Cytokine and chemokine values were determined in tears and correlated with the progression of keratoconus. Interestingly, we observed a significantly higher level of IL-1β, IL-10, IFN-γ, and TNF-α with moderate and severe keratoconus than with mild keratoconus. We also demonstrated a statistically significant positive correlation between TNF-α, IFN-γ, IL-6, and Ks and Kf values. Our study reveals alterations in cytokine concentrations in the tears of KC patients, supporting the hypothesis of inflammatory signalling in the pathway of the disease. The essential positive correlation between the three main pro-inflammatory cytokines and keratometry parameters allows us to conclude that the multifactorial aetiology of keratoconus appears to be even more complex, as inflammatory processes are probably also part of the pathophysiology of the disorder.

The analysis of Scheimpflug imaging parameters with tear inflammatory mediators in keratoconus was also performed by Pásztor et al. [41]. In their study, the significant associations were found between pairs of inflammatory mediators (IL-6 and CXCL8; CCL5 and CXCL8/MMP-9; TIMP-1 and MMP-9/-13/t-PA; t-PA and CXCL8/CCL5/PAI-1) and the severity of KC. In our study, we also observe a significant positive correlation between IL-6 and corneal parameters. According to our findings, inflammation appears to not only contribute to the pathogenesis of keratoconus but also plays a pivotal role in the pathological corneal processes throughout the entire spectrum, from the initial stages to the final ones. However, as we know, inflammatory mediators, including cytokines, take part in a complex cascade: inflammatory mediators overlap, neutralize and enhance each other’s effects. Thus, the study of only one or a few inflammatory mediators allows us to hypothesize the immunological basis of a given disease, but is not sufficient to investigate complex immunopathological processes.

Additionally, Ionescu et al. [42] found a strong statistically significant correlation between IL-6 concentration in tears and the degree of KC advancement (r = 0.56, *p* < 0.01), keratometry (r = 0.55, *p* < 0.02), pachymetry (r = −0.64, *p* < 0.048), and corneal hysteresis (*r* = −0.53, *p* < 0.02).

Fodor et al. [43] went a step further by comparing different degrees of advancement of keratoconus. They conducted research aiming to find immunomediator combinations which could sensitively indicate keratoconus progression. Tear samples of 42 patients with keratoconus were collected at baseline and at the end of a one-year follow-up. In their research, alterations in five Pentacam parameters exhibited a positive correlation with variations in IFN-γ, IL-13, IL-17A, CXCL8, CCL5, TIMP-1, and t-PA. Their findings indicate that the tear level of IL-13, in conjunction with NGF, can reliably predict the advancement of keratoconus, boasting a specificity of 100% and a sensitivity of 80%. The positive correlation of IFN-γ with Pentacam parameters is the common denominator of our studies.

In the long-term analysis, we observed a significantly positive correlation between IL-6 and TNF-α, IL-8/CXCL8, β-NGF, IFN-γ after 12 months following the CXL procedure. Similar results have been reported in publications by Kolozsvari et al. [19], confirming the hypothesis that cytokines and chemokines play an important role in the pathomechanism of keratoconus (KC), and that the CXL procedure may induce changes in the inflammatory response.

Many previous studies among KC patients after CXL have noted a significant improvement in corneal parameters. The results of our study are consistent with earlier findings. In the 12-month observation, we noted a statistically significantly lower Ks coefficient, Kf coefficient, PI-Apex, and corneal thickness. Therefore, the cornea in patients after CXL assumes a more regular shape, from this we find confirmation that previous research results affirm that CXL is an effective treatment for patients with progressive KC [19,44,45,46,47,48,49].

Moreover, Kobashi et al. [49] published a meta-analysis regarding the safety and effectiveness of the CXL procedure in paediatric patients with KC, revealing that all CXL techniques mitigated the progression of the disease in children with keratoconus for at least one year.

Despite some publications with different results [19,29], in our study, when comparing the concentrations of the cytokines we examined in the control group and the group with KC, we found that the levels of all the cytokines we investigated were significantly higher in the control group than in the study group. This may be a result of the tests used to detect the levels of these factors [50]. Our study was conducted based on results from flow cytometry, which is an extremely specialised technique used to differentiate cell populations based on morphological features and antigenic composition.

Our study has some limitations. The specific cellular origin of the examined cytokines remains ambiguous, as it is yet to be determined whether these bioactive molecules are produced by cells within the cornea, cells in the lacrimal glands, or a combination of both. Clarifying the cellular source is crucial for a comprehensive understanding of the underlying pathomechanisms of keratoconus and its potential causes.

## 4. Materials and Methods

### 4.1. Study Population and Measurements

A total of 52 patients were enrolled in this study, including 13 women and 39 men aged from 11 to 39, with an average age of 24.35, including 10 participants below the age of 18. The control group included 20 healthy people from whom tear samples were collected once. The control group consisted of healthy patients. 10 patients were women (50%) and 10 were men (50%). The mean age was 30 ± 3 years of age. The exclusion criteria for both patients with keratoconus and the control group were as follows: the existence of systemic diseases, individuals with dry eye disease (DED), the use of systemic and topical medications, a history of prior eye injuries and/or surgeries, ongoing eye inflammation, allergies in medical history, contact lens usage, pregnancy, and breastfeeding. These stringent criteria for selecting both the control and study groups were implemented to foster the creation of highly homogeneous groups, with the objective of minimizing potential complications in our research outcomes. In accordance with the tenets of the Declaration of Helsinki, all participants signed written informed consent prior to enrolment. This study has been accepted by the Bioethical Committee of Silesian Medical University—agreement no. PCN/0022/KB1/21/21.

Participants who had been diagnosed with progressive KC of various degrees and who underwent CXL surgery at the Ophthalmology Department of the Railway Hospital in Katowice between January 2020 and December 2022 were randomly recruited. All patients agreed to participate in this study. The criteria for assessing disease progression encompassed modifications in corneal topographic parameters. These changes included an escalation in astigmatism or corneal curvature (K1, K2), a Kmax increase of more than 1D within 12 months from the last follow-up visit, a posterior corneal elevation surge exceeding 15 µm, and a decline in visual acuity by one or more Snellen lines in corrected visual acuity. Comprehensive ophthalmic examinations were conducted for all patients, and tear samples were collected before and after CXL during a one-year observation period at regular intervals: 1 day before and after the surgery, at the day 7 visit, and at 1, 3, 6, 9, and 12 months after CXL. Moreover, all patients underwent a thorough ophthalmic examination, which included assessments of best-corrected visual acuity, slit-lamp examination, intraocular pressure measurement, and corneal topography using anterior segment optical coherence spectral tomography (SS OCT; CASIA2 OCT; Tomey, Nagoya, Japan). Evaluations encompassed flat (K 1), steep (K 2), cylindrical (CYL), and central corneal thickness (CCT) keratometry. Corneal sensitivity was quantified with a Cochet-Bonnet esthesiometer (Luneau Ophthalmologie, Paris, France), and corneal nerves were visualized using in vivo confocal microscopy (HRT 3, Heidelberg Engineering GmbH, Heidelberg, Germany).

The severity of keratoconus was categorized as follows: mild if the steepest keratometric reading (Ks) was <48 dioptres (D), moderate if Ks ranged from 48 to 54 D, and severe if Ks exceeded 54 D.

### 4.2. Cross-Linking Treatment

The CXL procedure adhered to the Dresden protocol and was conducted at the University Hospital Ophthalmology Department of the Railway Hospital in Katowice. Patients deemed eligible for the procedure had a corneal thickness exceeding 400 µm, making Epi-off (3 mW/cm^2^, 30 min) the preferred treatment. Post-procedure, a single drop each of levofloxacin (5 mg/mL) and dexamethasone (1 mg/mL) was administered, and a soft bandage contact lens (Air Optix Aqua; Ciba Vision, Alcon, Fort Worth, TX, USA, diameter 14.2, curvature 8.6) was applied. Typically, the contact lens was removed after the corneal epithelium had healed, usually at the initial follow-up visit (7 days post-procedure). Until the first follow-up visit, all patients utilized levofloxacin with dexamethasone eye drops and preservative-free artificial tears (containing sodium hyaluronate) five times daily. Subsequently, the antibiotic drops were gradually tapered and used for two weeks, while the steroid eye drops were continued for 1.5 months following the procedure.

### 4.3. Collection and Analysis of Tears

Tears of approximately 10 µL were non-invasively collected from the patients from both the study and control groups using micro-capillaries from the lower tear sac after gently folding the lower eyelid of the eye selected for the procedure. No local anaesthesia was used for this procedure. The collected tears were transferred to an Eppendorf tube and immediately stored in a freezer at −80 °C until the tears were analysed. The collected material samples had an average of approximately 10 µL in volume. Tear analysis using the Bio-Plex technique was performed at the Department of Microbiology and Immunology in Zabrze. The thawed tears were centrifuged at 4 °C at 14,000× *g* for 5 min. After centrifugation, the supernatants were transferred to “96F-well microplate” wells and diluted with a buffer to a volume of 50 µL. Standards of the analytes to be determined were also placed in the wells of the plate. Then, a suspension of magnetic beads coated with specific antibodies was added to the analytes being determined.

Further proceedings were in accordance with the manufacturer’s instructions. Finally, the obtained beads, coated with complexes and marked with phycoerythrin, were analysed using the Bio-Plex 3D Suspension Array System and xPONENT 4.0 acquisition software (Bio-Rad Laboratories Inc., Hercules, CA, USA). The concentrations of the measured cytokines were determined based on standard curves using the manufacturer’s software. The tear film multiplexes used test presence of cytokines (IL-1β, IL-1 RI, IL-6, IL-7, IL-8, IL-10, IL-12 IL-13, IL-15, EGF, KGF, IGF, FGF, HGF, PDGF, TNF-α, TGF-β, NGFβ, IFN-γ) and chemokines (CCL2/JE/MCP-1, CCL3/MIP-1α, CCL4/MIP-1β, and CXCL10/IP-10/CRG-2) [51,52].

### 4.4. Statistical Analysis

The Mann–Whitney test was used for comparisons of quantitative variables between the study group and the control group. For the correlation between reagents and the severity of keratoconus over time, the Kruskal–Wallis test was used. Spearman’s correlation coefficient was used to assess correlation between parameters and correlation of reagents with corneal parameters. Linear mixed models were used for the analysis of evolution of corneal parameters and the reagent’s values in time. The significance level was set to 0.05. All the analyses were conducted in R software, version 4.3.1.

## 5. Conclusions

Our results have shown that various processes related to immunological imbalance may contribute to the pathophysiology of keratoconus (KC). Furthermore, they provide a useful tool for monitoring the severity of the disease and predicting its progression.

All examined inflammatory mediators present in the tears film may play role in the development of KC, but their exact role is still unclear. Long-term continuous monitoring of cytokine levels correlated with KC progression measurements may establish the roles of particular factors present in disease development.

However, further research is necessary to prevent vision loss caused by this condition. This is especially important since the primary group affected by KC consists of young people, whose vision disorders often lead to visual disability, excluding them from society. According to our knowledge, this is the largest study analysing cytokine and chemokine levels immediately after the CXL procedure, in the tear film of patients with keratoconus and monitoring their concentrations over a year.

## Figures and Tables

**Figure 1 ijms-25-01052-f001:**
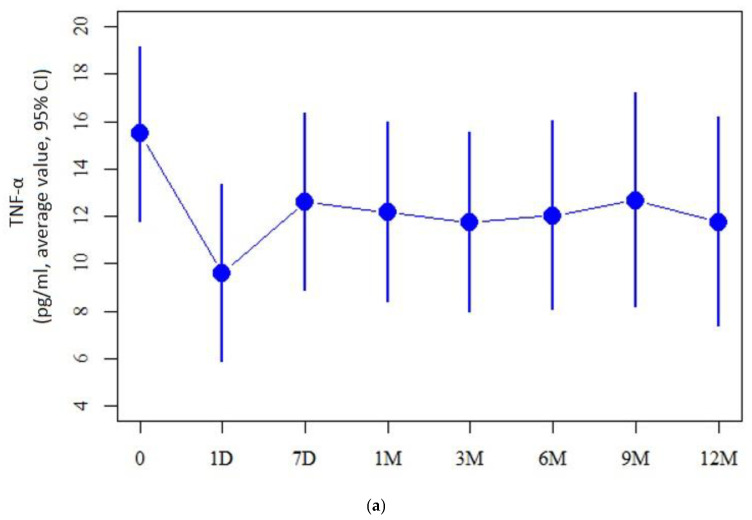
Cytokine level analysis of (**a**) TNF-α, (**b**) IL-1β, and (**c**) IL-6 was conducted in KC patients before CXL treatment (0), one day after CXL (1D), 7 days after CXL (7D), one month after CXL (1M), three months after CXL (3M), six months after CXL (6M), nine months after CXL (9M), and twelve months after CXL (12M).

**Table 1 ijms-25-01052-t001:** Comparison of pre- and post-CXL values of KC patient’s topographic measurements. The patient’s examinations were performed 1 day before the CXL procedure (0) and then repeated 7 days (7D) and 1 (1M), 3 (3M), 6 (6M), 9 (9M) and 12 (12M) months after the procedure. Ks—steep keratometric value; Kf—flat keratometric value; CYL—astigmatism value; PI-Apex—pachymetric index in the corneal apex; PI-Thinnest—pachymetric index in the thinnest corneal point.

Parameter	0	7D	1M	3M	6M	9M	12M
Ks [D]	49.8 (3.7)	50.8 (4.1)	50.3 (3.9)	49.4 (3.9)	49.7 (4.1)	49.8 (4.5)	48.5 (3.2)
Kf [D]	46.3 (3.2)	46.9 (3.6)	46.6 (3.6)	45.8 (3.4)	46.3 (3.8)	46.2 (4)	45.3 (2.9)
CYL [D]	3.5 (1.7)	4 (1.5)	3.8 (2)	3.6 (1.7)	3.4 (1.4)	3.6 (1.6)	3.2 (1.7)
PI-Apex [μm]	483 (30.8)	480.3 (59.9)	465.3 (37.7)	463.4 (49.3)	452.6 (36.4)	462 (39.9)	460.9 (40.8)
PI-Thinnest [μm]	451.4 (35.7)	450.7 (55.9)	436.6 (34.7)	429.9 (38.3)	421.1 (36.9)	435.3 (42.9)	429.2 (39)

**Table 2 ijms-25-01052-t002:** Correlation of corneal parameters with cytokine and chemokine level (**a**) pre-op, (**b**) 12 months after CXL. r—Spearman’s correlation coefficient; * statistically significant relationship (*p* < 0.05).

**(a)**					
	**Ks**	**Kf**	**Cyl**	**PI-Apex**	**PI-Thinnest**
TNF-α	r = 0.35, *p* = 0.017 *	r = 0.4, *p* = 0.006 *	r = 0.133, *p* = 0.379	r = −0.11, *p* = 0.467	r = −0.215, *p* = 0.151
IL-6	r = 0.339, *p* = 0.021 *	r = 0.447, *p* = 0.002 *	r = −0.09, *p* = 0.554	r = −0.13, *p* = 0.39	r = −0.185, *p* = 0.22
PDGF-BB	r = 0.227, *p* = 0.129	r = 0.305, *p* = 0.039 *	r = 0.043, *p* = 0.775	r = −0.199, *p* = 0.185	r = −0.282, *p* = 0.058
IL-8/CXCL-8	r = 0.316, *p* = 0.032 *	r = 0.353, *p* = 0.016 *	r = 0.124, *p* = 0.412	r = −0.142, *p* = 0.345	r = −0.058, *p* = 0.702
IL-7	r = 0.002, *p* = 0.99	r = −0.115, *p* = 0.448	r = 0.166, *p* = 0.269	r = −0.036, *p* = 0.813	r = 0.117, *p* = 0.438
CXCL-10/IP-10/CRG2	r = 0.227, *p* = 0.129	r = 0.286, *p* = 0.054	r = −0.014, *p* = 0.926	r = −0.229, *p* = 0.126	r = −0.326, *p* = 0.027 *
IL-10	r = 0.329, *p* = 0.026 *	r = 0.408, *p* = 0.005 *	r = 0.046, *p* = 0.76	r = −0.248, *p* = 0.096	r = −0.37, *p* = 0.011 *
CCL2/JE/MCP-1	r = 0.098, *p* = 0.519	r = 0.193, *p* = 0.199	r = −0.084, *p* = 0.58	r = −0.197, *p* = 0.19	r = −0.232, *p* = 0.121
NGF-β	r = 0.265, *p* = 0.075	r = 0.315, *p* = 0.033 *	r = 0.08, *p* = 0.597	r = −0.172, *p* = 0.254	r = −0.222, *p* = 0.138
IFN-γ	r = 0.244, *p* = 0.103	r = 0.331, *p* = 0.025 *	r = 0.021, *p* = 0.891	r = −0.22, *p* = 0.141	r = −0.239, *p* = 0.11
CCL-3/MIP-1a	r = 0.292, *p* = 0.049 *	r = 0.368, *p* = 0.012 *	r = 0.108, *p* = 0.474	r = −0.206, *p* = 0.17	r = −0.309, *p* = 0.037 *
CCL4/MIP-1b	r = 0.271, *p* = 0.068	r = 0.331, *p* = 0.024 *	r = 0.119, *p* = 0.431	r = −0.083, *p* = 0.586	r = −0.152, *p* = 0.313
FGFbasic/FGF2/bFGF	r = 0.316, *p* = 0.032 *	r = 0.354, *p* = 0.016 *	r = 0.059, *p* = 0.698	r = −0.04, *p* = 0.792	r = −0.232, *p* = 0.121
IL-15	r = 0.184, *p* = 0.221	r = 0.325, *p* = 0.028 *	r = −0.101, *p* = 0.505	r = −0.215, *p* = 0.151	r = −0.313, *p* = 0.034 *
IL-1β/IL-1F2	r = 0.309, *p* = 0.037 *	r = 0.315, *p* = 0.033 *	r = 0.149, *p* = 0.324	r = −0.154, *p* = 0.306	r = −0.253, *p* = 0.09
IL-1RI	r = 0.172, *p* = 0.252	r = 0.152, *p* = 0.314	r = 0.112, *p* = 0.459	r = −0.038, *p* = 0.8	r = −0.034, *p* = 0.821
IL-13	r = 0.248, *p* = 0.096	r = 0.325, *p* = 0.027 *	r = 0.04, *p* = 0.794	r = −0.064, *p* = 0.672	r = −0.172, *p* = 0.252
IL-12/IL-23 p40	r = 0.304, *p* = 0.04 *	r = 0.322, *p* = 0.029 *	r = 0.138, *p* = 0.361	r = −0.183, *p* = 0.222	r = −0.301, *p* = 0.042 *
**(b)**					
	**Ks**	**Kf**	**Cyl**	**PI-Apex**	**PI-Thinnest**
TNF-α	r = 0.583, *p* = 0.002 *	r = 0.593, *p* = 0.002 *	r = 0.165, *p* = 0.431	r = −0.338, *p* = 0.099	r = −0.274, *p* = 0186
IL-6	r = 0.455, *p* = 0.022 *	r = 0.423, *p* = 0.035 *	r = 0.162, *p* = 0.438	r = −0.45, *p* = 0.024 *	r = −0.373, *p* = 0.066
PDGF-BB	r = −0.081, *p* = 0.7	r = −0.05, *p* = 0.812	r = −0.082, *p* = 0.698	r = −0.066, *p* = 0.755	r = −0.021, *p* = 0.92
IL-8/CXCL-8	r = 0.201, *p* = 0.334	r = 0.084, *p* = 0.689	r = 0.187, *p* = 0.369	r = −0.099, *p* = 0.638	r = −0.029, *p* = 0.891
IL-7	r = −0.179, *p* = 0.39	r = −0.127, *p* = 0.544	r = −0.041, *p* = 0.847	r = −0.1, *p* = 0.633	r = 0.179, *p* = 0.391
CXCL-10/IP-10/CRG2	r = 0.232, *p* = 0.264	r = 0.347, *p* = 0.09	r = −0.237, *p* = 0.253	r = −0.179, *p* = 0.391	r = −0.294, *p* = 0.154
IL-10	r = 0.318, *p* = 0.122	r = 0.391, *p* = 0.054	r = −0.01, *p* = 0.963	r = −0.229, *p* = 0.271	r = −0.242, *p* = 0.244
CCL2/JE/MCP-1	r = 0.19, *p* = 0.361	r = 0.118, *p* = 0.574	r = 0.077, *p* = 0.714	r = 0.168, *p* = 0.423	r = 0.049, *p* = 0.815
NGF-β	r = 0.493, *p* = 0.012 *	r = 0.525, *p* = 0.007 *	r = 0.143, *p* = 0.495	r = −0.201, *p* = 0.335	r = −0.086, *p* = 0.681
IFN-γ	r = 0.664, *p* < 0.001 *	r = 0.564, *p* = 0.003 *	r = 0.247, *p* = 0.235	r = −0.367, *p* = 0.071	r = −0.262, *p* = 0.206
CCL-3/MIP-1a	r = 0.576, *p* = 0.003 *	r = 0.583, *p* = 0.002 *	r = 0.14, *p* = 0.505	r = −0.441, *p* = 0.027 *	r = −0.404, *p* = 0.045 *
CCL4/MIP-1b	r = 0.432, *p* = 0.031 *	r = 0.448, *p* = 0.025 *	r = 0.094, *p* = 0.654	r = −0.156, *p* = 0.457	r = −0.043, *p* = 0.836
FGFbasic/FGF2/bFGF	r = 0.445, *p* = 0.026 *	r = 0.504, *p* = 0.01 *	r = 0.01, *p* = 0.962	r = −0.449, *p* = 0.024 *	r = −0.411, *p* = 0.041 *
IL-15	r = 0.284, *p* = 0.17	r = 0.414, *p* = 0.04 *	r = −0.14, *p* = 0.503	r = −0.614, *p* = 0.001 *	r = −0.55, *p* = 0.004 *
IL-1β/IL-1F2	r = 0.421, *p* = 0.036 *	r = 0.402, *p* = 0.046 *	r = 0.053, *p* = 0.801	r = −0.054, *p* = 0.799	r = −0.022, *p* = 0.916
IL-1RI	r = 0.023, *p* = 0.914	r = −0.073, *p* = 0.728	r = 0.176, *p* = 0.398	r = 0.283, *p* = 0.17	r = 0.192, *p* = 0.359
IL-13	r = 0.569, *p* = 0.003 *	r = 0.564, *p* = 0.003 *	r = 0.228, *p* = 0.272	r = −0.172, *p* = 0.41	r = −0.199, *p* = 0.34
IL-12/IL-23 p40	r = 0.392, *p* = 0.052	r = 0.459, *p* = 0.021 *	r = −0.025, *p* = 0.907	r = −0.252, *p* = 0.224	r = −0.325, *p* = 0.113

## Data Availability

The data used to support the findings of this study are included in the article. The data will not be shared due to third-party rights and commercial confidentiality.

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
