# Peer review of "Analysis of Cytokine and Chemokine Level in Tear Film in Keratoconus Patients before and after Corneal Cross-Linking (CXL) Treatment"

_ijms, 2024, doi:10.3390/ijms25021052_

Round 1

Reviewer 1 Report

Comments and Suggestions for Authors

Dear Author

Thank you for your manuscript submission. The manuscript is well-presented and well-designed. The manuscript can be published in present form.

1. What is the main question addressed by the research?   The aim of the present study was to analyse the expressions of cytokines and chemokines in KC patients before CXL treatment and examine the potential changes in the concentration of these biomarkers at specified time intervals after the intervention.   2. Do you consider the topic original or relevant in the field? Does it address a specific gap in the field?   Yes, Yes   3. What does it add to the subject area compared with other published material?   The manuscript study presents effective Results in the field with rigorous and precise methodology   4. What specific improvements should the authors consider regarding the methodology? What further controls should be considered?   None-None   5. Are the conclusions consistent with the evidence and arguments presented and do they address the main question posed?   Yes   6. Are the references appropriate?   Yes   7. Please include any additional comments on the tables and figures.
None

Reviewer 2 Report

Comments and Suggestions for Authors

The authors showed impressive study that they analyzed the expressions of cytokines and chemokines in KC patients before and after specified time intervals after CXL treatment.

The authors showed that various processes related to immunological imbalance play a role in the pathophysiology of Keratoconus. I have a few minor comments to be addressed.

As authors demonstrated a statistically significant positive correlation between TNF-α, IFN-γ, IL-6, and Ks and Kf values. These correlations are very impressive results in the manuscript.

How did the authors think about the mechanism of Keratoconus? In discussion part, the authors showed a lot of references, but they didn’t describe their hypothesis about mechanism of Keratoconus by the authors data. My suggestion is to describe this part in discussion part.

Reviewer 3 Report

Comments and Suggestions for Authors

The current manuscript aims to analyze cytokine and chemokine level in tear film in keratoconus patients before and after corneal cross-linking (CXL) treatment. However, in my opinion, the article content is seriously limited by insufficient academic novelty and scientific progress. There are several similar reports on the specific topic already existed in the literature database. Please refer to the following example papers: #1 Balasubramanian et al. Proteases, proteolysis and inflammatory molecules in the tears of people with keratoconus. Acta Ophthalmol 2012;90:e303-309. & #2 Akçay et al. Tear function and ocular surface alterations after accelerated corneal collagen cross-linking in progressive keratoconus. Eye Contact Lens 2017;43:302-307. & #3 Uysal et al. Tear function and ocular surface changes following corneal collagen cross-linking treatment in keratoconus patients: 18-month results. Int Ophthalmol 2020;40:169-177. & #4 Eser et al. Evaluation of keratoconus disease with tear cytokine and chemokine levels before and after corneal cross-linking treatment. Ocul Immunol Inflamm 2023:1-7. Overall, according to my experience, this report seems difficult to attract more attention from the scholars working in the same field. Another major drawback is that the authors do not provide insightful viewpoints to make knowledge contribution. Based on above considerations, this reviewer cannot recommend that this particular article without significant scientific advancement of any aspect of “molecular sciences” is suitable for publication to this high-quality journal.

Reviewer 4 Report

Comments and Suggestions for Authors

The manuscript titled "Analysis of Cytokine and Chemokine Levels in Tear Film in Keratoconus Patients Before and After Corneal Cross-Linking (CXL) Treatment" presents the results of interesting and valuable research that enhances our understanding of the pathogenesis of keratoconus. This study represents a valuable effort in advancing our knowledge of this issue. However, I have a few remarks and suggestions.

In the abstract it is necessary to state at which time intervals the subjects' tears were collected. Furthermore, in the abstract, it is required to include data regarding the presence of a control group, along with detailing its basic characteristics. Additionally, ensure an explanation of the abbreviation “CXL” (corneal cross-linking) in the abstract.

It is essential to provide explanations for abbreviations when they are initially introduced in the text. Additionally, ensure that abbreviations are sufficiently explained in all tables the benefit of readers.

In the introduction section, it is advisable to outline the fundamental characteristics of the analysed cytokines and chemokines. Moreover, provide an explanation for the rationale behind their selection for analysis.

In the results section, include the basic demographic characteristics of individuals in the control group, their ophthalmological status, and the values of topographic measurements. Specify whether the analysis of cytokines and chemokines in tears was conducted for the control group throughout all time periods similar to the study participants, or solely at the beginning of the study.

In Table 1, it is essential to provide an explanation for the numbers in brackets to enhance reader comprehension.

Breastfeeding and pregnancy were listed as exclusion criteria twice: first as "Breastfeeding, pregnancy" and secondly as "Pregnant and breastfeeding women."

The initial ophthalmological examination was conducted one day before the procedure. It is essential to elucidate why the first tear samples were not collected at the same time as the ophthalmological examination was performed or some time before the procedure. If such samples were not obtained, the initial sample cannot be designated as preoperative.

In the discussion section, indicate whether there are similar studies in patients with keratoconus with a longer follow-up period. Specify whether, in studies correlating cytokine and chemokine values with keratoconus progression, these values were determined in serum or tears. Additionally, explore if there are studies where specific mediators in tears were assessed in patients with keratoconus, monitored for changes in their values with the progression of keratoconus, and describe the outcomes of these studies.Vrh obrasca

 The conclusion of the paper should be reformulated. Based on the observed postoperative changes, it can be hypothesized that the examined mediators in tears may contribute to the development of KC, but it is premature to assert that they play a key role. A definitive conclusion would necessitate continuous monitoring of cytokine levels and KC progression over an extended period and establishing their correlation.

Round 2

Reviewer 3 Report

Comments and Suggestions for Authors

The authors' revision is highly appreciated again. Although this reviewer has different novelty viewpoint from the authors' group, I respect their responses to my comments. Therefore, this reviewer changes the recommendation to support the publication of the article in the journal "IJMS".

Reviewer 4 Report

Comments and Suggestions for Authors

The authors have responded appropriately to the suggestions and have implemented all the requested changes, resulting in an improved quality of the manuscript. Therefore, the manuscript can be accepted in its current form.
